# CAML: Collaborative Auxiliary Modality Learning for Multi-Agent Systems

## Abstract

Multi-modality learning has become a crucial technique in enhancing the performance of machine learning applications across various domains, including autonomous driving, robotics, and perception systems. Existing frameworks, such as Auxiliary Modality Learning (AML), effectively utilize multiple data sources during training and enable inference with reduced modalities, but they primarily operate in a single-agent context. This limitation is particularly critical in dynamic environments, such as connected autonomous vehicles (CAV), where incomplete data coverage can result in decision-making blind spots. To address these challenges, we introduce Collaborative Auxiliary Modality Learning (**CAML**), a novel extension of the AML framework for multi-agent systems. **CAML** facilitates collaboration among agents by allowing them to share multimodal data during training. During inference, each agent operates effectively with fewer modalities, ensuring robustness in performance even with missing data. We analyze the effectiveness of **CAML** from the perspective of uncertainty reduction and data coverage, providing a theoretical support to understand and explain why **CAML** works better than AML. We then validate **CAML** through experiments in collaborative decision-making for CAV in accident-prone scenarios. Experimental results show that **CAML** outperforms AML across all tested scenarios, achieving up to a $58.3\%$ improvement in accident detection. Additionally, we validate our approach on real-world data from aerial-ground vehicles for collaborative semantic segmentation, achieving up to $10.8\%$ improvement in mIoU compared to AML.

## 1 Introduction

Multi-modality learning has become an essential approach in a wide range of machine learning applications, particularly in areas such as autonomous driving (El Madawi et al., 2019; Xiao et al., 2020; Gao et al., 2018) , robotics (Noda et al., 2014; Lee et al., 2020), and perception systems (Zhuang et al., 2021; Bayoudh et al., 2022), where the availability of multiple data sources (e.g., RGB images, LiDAR, radar, etc.) improves model performance by providing complementary information. However, these multi-modality systems often suffer from increased computational complexity and latency at inference time. Moreover, some modalities may not be consistently available or reliable in real-world conditions, necessitating strategies that can compensate for missing modalities during inference.

Recent work on machine learning (Hoffman et al., 2016; Wang et al., 2018; Garcia et al., 2018; 2019; Piasco et al., 2021) aims to address these problems by allowing models to leverage additional modalities during training while enabling inference using fewer or even a single modality. For example, a model might be trained using both RGB and LiDAR data, but during deployment, it only requires RGB data to operate. These approaches reduces the computational burden and accommodates real-world conditions where certain sensors may be unavailable. Shen et al. (2023) formalized these learning tasks as Auxiliary Modality Learning (AML). The AML framework successfully reduces the dependency on expensive or unreliable modalities, but it focuses on the single-agent setting, where an individual model is trained to handle reduced modalities during inference.

Despite the benefits of AML, several gaps remain. First, a major limitation in the current AML framework is the inability to exploit collaboration between agents, particularly in dynamic environments such as *connected autonomous vehicles* (CAV). In such scenarios, data coverage from a

single agent is often incomplete because of occlusion or limited sensor range, leading to blind spots or increased uncertainty in decision-making. Second, the information from multiple modalities can complement each other across agents, especially in multi-agent settings such as vehicle-to-vehicle (V2V) communication or collaborative robotics. Different agents may have access to complementary sensory information, which could be shared to have agents make more informed and safer decisions, notably in accident-prone scenarios. However, current AML approaches do not exploit this potential for collaboration.

To bridge these gaps, inspired by AML (Shen et al., 2023), we propose Collaborative Auxiliary Modality Learning (**CAML**), a significant extension of the AML framework to multi-agent systems, which is much more genreal than AML. **CAML** allows agents to collaborate and share multimodal data during training, but enables inference with reduced or fewer modalities per agent. As suggested by Shen et al. (2023), **CAML** leverages knowledge distillation (Hinton, 2015), transferring knowledge from a teacher model into a student model. This enables the student to operate with missing modalities during inference. For instance, in autonomous driving, multiple vehicles can share sensor information such as LiDAR and RGB images during training to build more robust representations, while during runtime testing, each vehicle performs inference using only RGB images.

**CAML** addresses two key challenges: First, it reduces uncertainty and enhances data coverage in dynamic environments by leveraging complementary information from multiple agents. Second, it maintains efficient, modality-reduced inference during testing. Unlike previous work that either focuses on multi-agent collaboration but without addressing modality reduction at test time, or tackles multi-modality learning in single-agent settings, **CAML** unifies these concepts. Through collaboration, **CAML** enables agents to compensate for each other's blind spots, resulting in more informed prediction or decision-making even when some modalities are unavailable at deployment.

In summary, our work offers the following key contributions:

- We introduce **CAML**, a novel framework that extends Auxiliary Modality Learning (AML) to multi-agent systems. **CAML** allows agents to share multimodal data during training, while performing efficient, reduced-modality inference during testing. By shifting AML from a single-agent paradigm to a collaborative framework, **CAML** leverages the strengths of multi-agent collaboration, reducing estimation uncertainty and offering complementary information to capture a broader and more detailed data representation.
- We systematically analyze the effectiveness of **CAML** from the perspective of uncertainty reduction and data coverage, providing a theoretical support to understand and explain why **CAML** works better than AML.
- We validate **CAML** through experiments in collaborative decision-making for connected autonomous driving in accident-prone scenarios and collaborative semantic segmentation for real-world data of aerial-ground vehicles. **CAML** demonstrates up to $58.3\%$ improvement in accident detection compared to AML in decision-making for autonomous driving, significantly enhancing driving safety, and up to $10.8\%$ improvement in mIoU for semantic segmentation.

## 2 RELATED WORK

**Multi-Agent Collaboration.** Collaboration in multi-agent systems has been widely studied across fields such as autonomous driving and robotics. In autonomous driving, prior research has explored various strategies, including spatio-temporal graph neural networks (Gao et al., 2024), LiDAR-based end-to-end systems (Cui et al., 2022), decentralized cooperative lane-changing (Nie et al., 2016) and game-theoretic models (Hang et al., 2021). In robotics, Mandi et al. (2024) presented a hierarchical multi-robot collaboration approach using large language models, while Zhou et al. (2022) proposed a perception framework for multi-robot systems built on graph neural networks. A review of multi-robot systems in search and rescue operations was provided by Queralta et al. (2020), and Bae et al. (2019) developed a reinforcement learning (RL) method for multi-robot path planning. Additionally, various communication mechanisms, such as Who2com (Liu et al., 2020b), When2com (Liu et al., 2020a), and Where2comm (Hu et al., 2022), have been created to optimize agent interactions.

Despite these advancements, existing multi-agent collaboration frameworks remain limited by their focus on specific tasks and the assumption that agents will have consistent access to the same data modalities during both training and testing, an assumption that may not hold in real-world applications. To address these gaps, our framework, **CAML**, enables agents to collaborate during training

by sharing multimodal data, but at test time, each agent performs inference using reduced modality. This reduces the dependency on certain modalities for deployment, while still allowing agents to leverage additional data during training to enhance overall performance and robustness.

**Auxiliary Modality Learning.** Auxiliary Modality Learning (AML) (Shen et al., 2023) has emerged as an effective solution to reduce computational costs and the amount of input data required for inference. By utilizing auxiliary modalities during training, AML minimizes reliance on those modalities at inference time. For example, Hoffman et al. (2016) introduced a method that incorporates depth images during training to enhance test-time RGB-only detection models. Similarly, Wang et al. (2018) proposed PM-GANs to learn a full-modal representation using data from partial modalities, while Garcia et al. (2018; 2019) developed approaches that use depth and RGB videos during training but rely solely on RGB data for testing. Piasco et al. (2021) created a localization system that predicts depth maps from RGB query images at test time. Building on these works, Shen et al. (2023) formalized the AML framework, systematically classifying auxiliary modality types and AML architectures.

However, existing AML frameworks are typically designed for single-agent settings, failing to exploit the potential benefits of multi-agent collaboration for improving multimodal learning. **CAML** extends AML to multi-agent environments, allowing agents to collaboratively learn richer multimodal representations during training. This approach mitigates the loss of information when modalities are reduced during inference, as the learned features are reinforced by data shared across agents.

**Knowledge Distillation.** Knowledge distillation (KD) (Hinton, 2015) is a widely used technique in many domains to reduce computation by transferring knowledge from a large, complex model (teacher) to a simpler model (student). In computer vision, (Gou et al., 2021) provided a comprehensive survey of KD applications, while Beyer et al. (2022) conducted an empirical investigation to develop a robust and effective recipe for making SOTA large-scale models more practical. Additionally, Tung & Mori (2019) introduced a KD loss function that aligns the training of a student network with input pairs producing similar activation in the teacher network. In natural language processing, Xu et al. (2024) reviewed the applications of KD in LLMs, while Sun et al. (2019) proposed a Patient KD method to compress larger models into lightweight counterparts that maintain effectiveness. Hahn & Choi (2019) also suggested a KD approach that leverages the soft target probabilities of the training model to train other neural networks. In autonomous driving, Lan & Tian (2022) presented an approach for visual detection, Cho et al. (2023); Sautier et al. (2022) used KD for 3D object detection.

Notice that existing KD mostly distills knowledge from a larger model to a smaller one to reduce computation, Shen et al. (2023) aimed to design a cross-modality learning approach using KD to utilize the hidden information from auxiliary modalities within the AML framework. But AML is limited by the scope of a single-agent paradigm, missing opportunities for collaborative knowledge sharing across agents. In contrast, we leverage KD within multi-agent settings, where the teacher models are trained with access to shared multimodal data (e.g., RGB and LidAR) from multiple agents. By distilling this collaborative knowledge into each agent's reduced modality (e.g., RGB), **CAML** enables robust inference during deployment, even with fewer modalities. This collaborative distillation process enhances each agent's performance by providing richer, complementary knowledge from the collaborative training phase.

## 3 Collaborative Auxiliary Modality Learning

In AML (Shen et al., 2023), which operates in a single-agent framework, the missing modalities during testing are referred to as auxiliary modalities, while those that remain available are called the main modality. In contrast, our framework **CAML** is more general for multi-agent collaboration. Each agent can process a different number of modalities during training, different agents in **CAML** can have different main modalities and auxiliary modalities. There is no correlation between the number of agents and the number of modalities.

We define our problem in both training and testing phases. In the training phase, we consider a multi-agent system with $N$ agents collaboratively completing a task. The set of agents is denoted as $\mathcal{A}_{train} = \{\mathcal{A}_1, \mathcal{A}_2, \ldots, \mathcal{A}_N\}$. The observations of all agents are denoted as $X = \{x_1, x_2, \ldots, x_N\}$, where $x_i$ is the observation acquired by the $i$-th agent $\mathcal{A}_i \in \mathcal{A}_{train}$. The ground truth label is

denoted as $Y$, which can be an object label, semantic class, or a control command (e.g., brake for an autonomous vehicle). The set of modalities is denoted as $\mathcal{I}_{train} = \{\mathcal{I}_1, \mathcal{I}_2, \ldots, \mathcal{I}_K\}$, such as RGB, LiDAR, Depth, etc, where $K$ is the number of modalities avaiable during training. During training, each agent has access to all these $K$ modalities.

In the testing phase, we assume there are $M$ agents. The set of test agents is denoted as $\mathcal{A}_{test} = \{\mathcal{A}_1, \mathcal{A}_2, \ldots, \mathcal{A}_M\}$. In addition, the set of modalities is denoted as $\mathcal{I}_{test}$, which is a subset of $\mathcal{I}_{train}$. The number of modalities available during testing is denoted as $L$, where $L \leq K$. The set of agents that have access to the $j$-th modality $\mathcal{I}_j \in \mathcal{I}_{test}$ is denoted as $\mathcal{A}_{test}^{\mathcal{I}_j}$, where $\mathcal{A}_{test}^{\mathcal{I}_j} \in \mathcal{A}_{test}$, and the number of agents in this set is given by $|\mathcal{A}_{test}^{\mathcal{I}_j}| = M_j$. This means that during testing, each agent may have access to different number of modalities.

Given the problem definition, we aim to estimate the posterior distribution $P(y|X)$ of the ground truth label $y$ given all agents' observations $X$. During training, we train both a teacher model where each agent has access to all modalities in $\mathcal{I}_{train}$ and a student model where each agent has access to partial modalities in $\mathcal{I}_{test}$. We employ Knowledge Distillation (KD) to transfer the knowledge derived from the teacher model to the student model, enabling the student to benefit from additional information, as illustrated in Figure 1. At test time, we perform inference using the student model, which relies on the test modality observations $X^{test}$.

Specifically, in the teacher model, each agent has access to all multimodal observations and independently processes its local observations to produce embeddings. These embeddings are then shared among agents based on whether the system operates in a centralized or decentralized manner. If the system is centralized, all collaborative agents share their embeddings with one designated ego agent for centralized processing. If the system is decentralized, each agent shares the embeddings with other agents. Subsequently, the shared embeddings corresponding to the same modality are aggregated together. Then we fuse (e.g., via concatenation) the aggregated embeddings of different modalities to create a comprehensive multimodal embedding. This multimodal embedding is then passed through a prediction module to produce the teacher model's final prediction. The student model follows a similar network architecture as the teacher. However, instead of processing all modalities, each agent processes only a single or a subset of modalities, which can vary across agents. By sharing these embeddings among agents, the student model also constructs a multimodal embedding, leveraging the different modalities observed by various agents. This multimodal embedding is then used to generate the student model's prediction. Thus, our approach enables the student model to maintain strong predictive performance despite missing modalities during testing, significantly enhancing its robustness and generalizability.

## 4 ANALYSIS

To compare whether **CAML** outperforms AML with a single agent theoretically, we analyze from two key perspectives: *uncertainty reduction* and *data coverage enhancement*. Data coverage can be further discussed from two dimensions: complementary information and information gain. We aim to address three major questions: (a) **Uncertainty Reduction**: Does the collaboration among multiple agents help reduce the variance of the posterior distribution, resulting in more confident estimates? (b) **Complementary Information**: Does the collaboration of multiple agents provide complementary information that increases data coverage? Specifically, does combining observations from each agent lead to a more accurate and comprehensive prediction compared to using a single agent? (c) **Information Gain**: Does the collaboration increase the mutual information between the observations and the true label?

**Uncertainty Reduction.** To address question (a) about uncertainty reduction, the prior $P(y)$ is typically assumed to be Gaussian: $P(y) = \mathcal{N}(y|\mu_0, \sigma_0^2)$, where $\mu_0$ is the prior mean and $\sigma_0^2$ is the prior variance.

**Single-Agent**. In the single-agent case, we assume that only agent $\mathcal{A}_i$ is available and its likelihood $P(x_i|y)$ is Gaussian: $P(x_i|y) = \mathcal{N}(x_i|\mu_i(y), \sigma_i^2)$, where $\mu_i(y)$ is the mean of the observation $x_i$ given $y$, $\sigma_i^2$ is the variance of the agent $\mathcal{A}_i$'s observations. The posterior distribution $P(y|x_i)$ is proportional to the product of the prior and likelihood, $P(y|x_i) \propto P(y)P(x_i|y)$, which is also Gaussian. And the posterior variance $\sigma_{single}^2 = \left(\frac{1}{\sigma_0^2} + \frac{1}{\sigma_i^2}\right)^{-1}$ (Murphy, 2007).

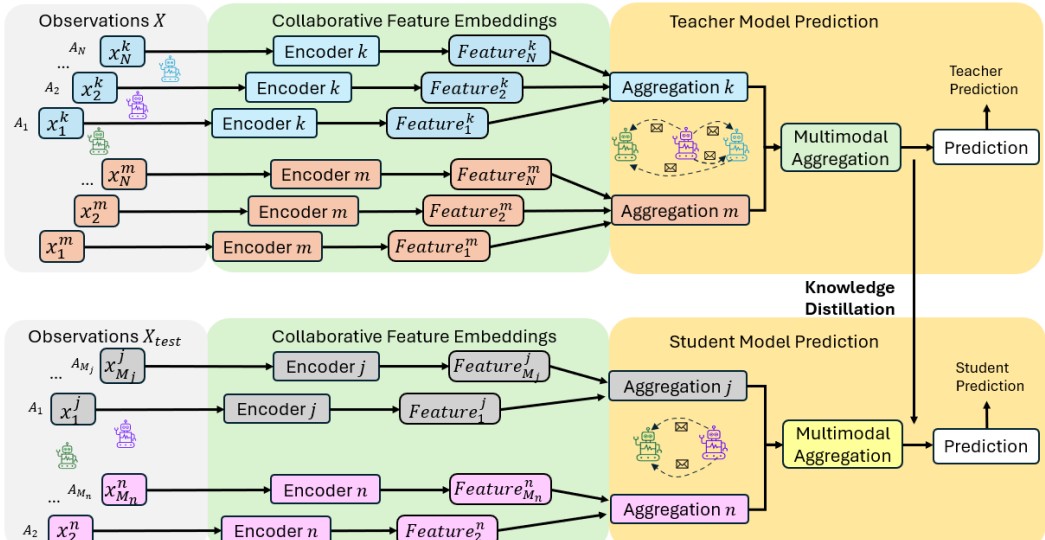

Figure 1: **CAML Approach Pipeline.** The teacher model (top) aggregates and shares multimodal embeddings across agents for prediction. In contrast, the student model (bottom) processes a subset of modalities per agent and shares them to form a multimodal embedding. This allows the student model to handle missing modalities during testing, while still generating robust predictions. Please see details in Section 3.

**Multi-Agent**. In the case of multi-agent collaboration, we model the joint likelihood of the observations $X$ as a multivariate Gaussian distribution, conditioned on the true target variable $y$: $P(X|y) = \mathcal{N}(X|\mu_X(y), \Sigma)$, where $\mu_X(y)$ is the joint mean of the observations from all agents, conditioned on $y$, $\Sigma$ is the covariance matrix, encoding the correlations between the observations from multiple agents. The posterior $P(y|X) \propto P(y)P(X|y)$, is another Gaussian, with variance $\sigma^2_{multi} = \left(\frac{1}{\sigma_0^2} + \mathbf{1^T\Sigma^{-1}1}\right)^{-1}$ (Murphy, 2007).

Since $\mathbf{1^T\Sigma^{-1}1} \geq \frac{1}{\sigma_i^2}$ for any $i$, we have $\sigma^2_{multi} \leq \sigma^2_{single}$. In the extreme case where all agents' observations are perfectly correlated (e.g., they all observe the same thing), the posterior variance would be equivalent to that of a single agent. However, multi-agent collaboration reduces variance compared to a single agent, as long as the observations are not perfectly correlated, proving that collaboration reduces uncertainty.

**Data Coverage.** When comparing data coverage between **CAML** and AML, we analyze it from two key dimensions: complementary information and information gain.

**Complementary Information.** To address question (b) about complementary information, we study data coverage and information provided by each agent in a multi-agent system. Let the entire data space be denoted as $\mathcal{D}$, which consists of various subsets. Each agent $\mathcal{A}_i$ in the system covers a subset of this data space: $\mathcal{C}_i \subseteq \mathcal{D}$. The overall coverage by the system is given by the union of all subsets covered by individual agents: $\mathcal{C}_{multi} = \cup_{i=1}^{N}\mathcal{C}_i$. This ensures that $|\mathcal{C}_{multi}| \geq \max |\mathcal{C}_i|$. If only a single agent is available, it can only observe a portion of the data space, leaving parts of the space unobserved, which leads to incomplete information for estimating the true label $y$. We show an qualitative example of multi-agent collaboration provides complementary information to enhance data coverage in Fig . 6 in the Appendix.

From a probabilistic perspective, when multi-agent collaboration is in place, the combined likelihood $P(X|y)$ is modeled as a multivariate distribution (as discussed in Section 4). This approach provides a broader and more accurate representation of the data space by integrating information from all agents and modeling the dependencies and correlations between them. Compared to a univariate distribution $P(x_i|y)$ for a single agent $\mathcal{A}_i$, the multivariate distribution covers a larger portion of the data space $\mathcal{D}$, thus enhancing data coverage. This allows the exploration of more complex patterns, relationships, and complementary information from different agents. By capturing a richer set of interactions and correlations among the agents' observations, the multivariate distribution sup-

ports more informed decision-making. The model's predictions are based on a comprehensive view of the environment, thus leading to more accurate outcomes.

**Information Gain.** To address question (c) about information gain, we analyze using information theory. Let $I(y; x_i)$ represent the mutual information between the true label $y$ and agent $\mathcal{A}_i$'s observation $x_i$, which quantifies how much information $x_i$ provides about the estimation of $y$. The mutual information between $y$ and the set of all observations $X$ is $I(y; X)$.

In the context of multi-agent collaboration, the joint observations $X$ from multiple agents typically provide more comprehensive information about the true label $y$ compared to the observation of any single agent. Therefore, the mutual information $I(y; X)$ is always greater than or equal to the mutual information from a single agent: $I(y; X) \geq I(y; x_i)$. Thus, the combined observations from multi-agent collaboration provide more information about $y$ than a single observation, improving the overall estimate. By leveraging the combined knowledge from multiple agents, the prediction of $y$ becomes more accurate, reflecting the added value of collaboration. Multi-agent systems are generally more informative, as the interaction and joint information between agents can reduce uncertainty about the target variable, as discussed in Section 4.

## 5 EXPERIMENTS

### 5.1 COLLABORATIVE DECISION-MAKING IN CONNECTED AUTONOMOUS DRIVING

To evaluate our approach, we first focus on collaborative decision-making in connected autonomous driving (CAV). This involves making critical decisions for the ego vehicle in accident-prone scenarios, such as determining whether or not to take a braking action.

**Data Collection.** We utilize a connected autonomous driving simulation that integrates CARLA (Dosovitskiy et al., 2017) with AutoCast (Qiu et al., 2021). Following prior research (Cui et al., 2022; Gao et al., 2024), we focus on three complex traffic scenarios prone to accidents due to limited sensor coverage or obstructed views, as illustrated in Fig. 2: (1) Overtaking: A sedan is blocked by a truck on a narrow, two-way road with a dashed centerline. The truck also obscures the sedan's view of oncoming traffic. The ego vehicle must decide when and how to safely pass the truck. (2) Left Turn: The ego vehicle attempts a left turn at a yield sign. Its view is partially blocked by a truck waiting in the opposite left-turn lane, reducing visibility of vehicles coming from the opposite direction. (3) Red Light Violation: As the ego vehicle crosses an intersection, another vehicle runs a red light. Due to nearby vehicles waiting to turn left, the ego vehicle's sensors are unable to detect the violator.

For each scenario, we collect 24 trials, dividing them into 12 trials for training through behavior cloning (BC) and 12 trials for testing. Each trial includes RGB images and LiDAR point clouds captured by a variable number of connected vehicles, along with the ground truth actions of the ego vehicle. At each timestamp, the ego vehicle has a maximum of three collaborative vehicles, provided their distance is within a threshold of 150 meters (Gao et al., 2024; Cui et al., 2022). For each vehicle, both RGB and LiDAR data are used during training, while only RGB data is used during testing in **CAML**.

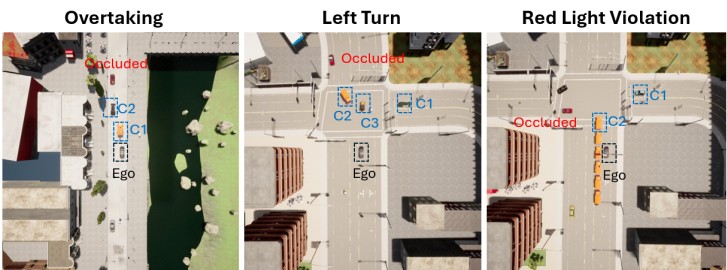

Figure 2: Three accident-prone scenarios in connected autonomous driving: overtaking, left turn, and red light violation.

**Experimental Setup.** For processing RGB data, we first resize the image to $224 \times 224$ and use ResNet-18 (He et al., 2016) as the encoder to extract a feature map. We then apply self-attention on the feature map to dynamically compute the importance of features at different locations. After the

self-attention, we apply three convolution layers with each followed by a ReLU activation. Finally, we obtain a 256-dimensional feature representation after passing through a fully connected layer. To facilitate the collaboration and aggregation of RGB feature embeddings from connected vehicles to the ego vehicle, we use the cross-attention mechanism. For processing the LiDAR data, we use the Point Transformer (Zhao et al., 2021) as the encoder and utilize the COOPERNAUT (Cui et al., 2022) model to aggregate LiDAR feature embeddings.

For the training of Knowledge Distillation (KD), we first train a teacher model offline using a binary cross-entropy loss, where each vehicle has both RGB and LiDAR data. Then we train a student model to mimic the behavior of the teacher model with only RGB data for each vehicle. For each data point, the student model receives the same RGB image that the teacher model was given. For further details on the KD training process, please refer to Appendix A.4. For the prediction module, we use a three-layer MLP. And for the detailed training settings, please see Appendix A.3.1.

We employ the following two metrics for evaluation: (1) **Accident Detection Rate (ADR)**: This is the ratio of accident-prone scenarios correctly detected by the model compared to the total ground truth accident-prone scenarios. An accident-prone case is identified when the ego vehicle performs a braking action. This metric measures the model's effectiveness in identifying potential accidents. (2) **Expert Imitation Rate (EIR)**: This denotes the percentage of actions accurately replicated by the model out of the total expert actions. It serves to evaluate how well the model mimics expert driving behavior.

**Baselines.** We implement the following three baselines for comparison: (1) **AML** (Shen et al., 2023): In the AML setting, the ego vehicle operates independently without collaboration with other vehicles (non-collaborative). Both RGB and LiDAR data are available during training for the vehicle, while only RGB data is available during testing. (2) **COOPERNAUT** (Cui et al., 2022) (Single-Agent): Processes LiDAR data during both training and testing. COOPERNAUT uses the Point Transformer (Zhao et al., 2021) as the backbone, encoding raw 3D point clouds into keypoints. (3) **STGN** (Gao et al., 2024) (Single-Agent): Utilizes spatial-temporal graph networks for decision-making, with RGBD data used for both training and testing.

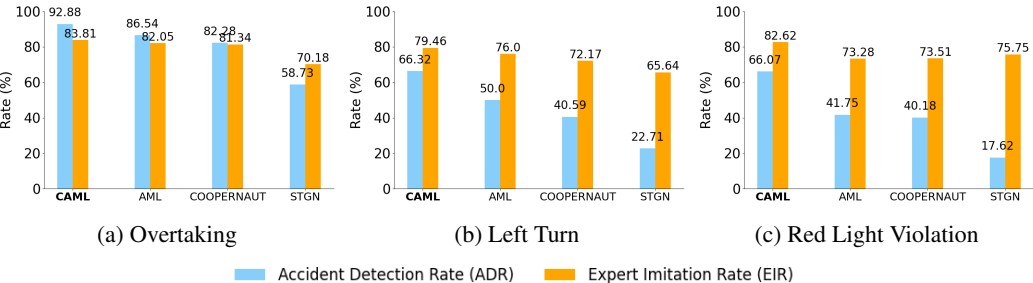

(a) Overtaking    (b) Left Turn    (c) Red Light Violation

Figure 3: **Performance Comparison of CAML Against Baselines.** We evaluate performance using two metrics: Accident Detection Rate (ADR) and Expert Imitation Rate (EIR) across three accident-prone scenarios: (a) Overtaking, (b) Left Turn, and (c) Red Light Violation. The baselines, AML, COOPERNAUT, and STGN, operate in a single-vehicle, non-collaborative setting. In contrast, *CAML demonstrates superior performance across all scenarios compared to these baselines, benefiting from the multi-agent collaboration.*

**Baseline Comparison.** How well does **CAML** perform against other methods for decision-making in CAV? We evaluate **CAML** against the baselines and present the results in Figure 3, which demonstrate a clear performance advantage of **CAML** across all three accident-prone scenarios. The evaluation metrics, accident detection rate (ADR) and expert imitation rate (EIR), reveal that **CAML** consistently outperforms AML, COOPERNAUT, and STGN. In particular, **CAML** achieves notable improvements in ADR compared to AML: $13.2\%$ in the overtaking scenario, $32.6\%$ in the left turn scenario, and a significant $58.3\%$ in the red light violation scenario. The more pronounced improvements in the left turn and red light violation scenarios can be attributed to the higher complexity of these situations, where restricted views and occlusions present greater challenges for decision-making. Unlike the overtaking scenario on a two-way road, which is relatively less constrained, left turns and red light violations often involve more unpredictable vehicle and pedestrian interactions, requiring enhanced situational awareness. In these more demanding cases, the collaborative framework of **CAML** proves especially beneficial, as it allows the ego vehicle to aggregate

additional sensory data from connected vehicles, significantly boosting its capacity to detect potential accidents and respond proactively, such as applying braking when necessary to avoid collisions.

As detailed in Section 4, the collaborative nature of **CAML** plays a critical role in reducing the uncertainty in the decision-making processes. By incorporating sensory data from multiple connected vehicles, **CAML** can draw on a richer and more diverse dataset, which enables more reliable predictions. This collaborative approach not only reduces the uncertainty in estimations but also enhances data coverage by leveraging complementary information from all connected agents. As a result, the ego vehicle is able to form a more accurate and comprehensive understanding of its environment, particularly in scenarios where its own sensing capabilities are limited by obstructions or blind spots. Compared to single-agent systems, where decisions rely solely on local sensory data, the multi-agent collaboration in **CAML** allows the ego vehicle to better handle complex driving environments, especially in accident-prone situations. These baseline comparison results of improvements in safety and decision-making align well with our theoretical analysis.

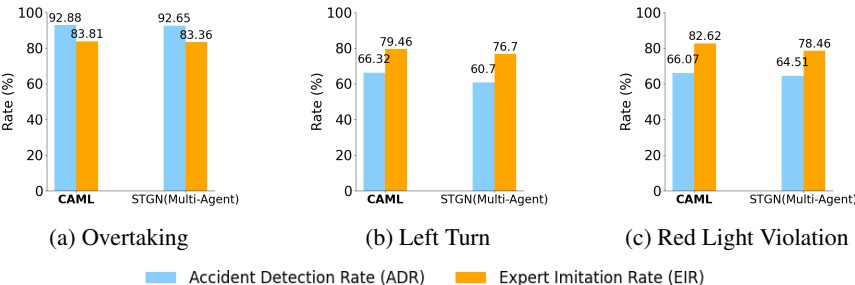

(a) Overtaking      (b) Left Turn      (c) Red Light Violation

Figure 4: **Modality-Efficient Superiority of CAML Against STGN with Multi-Agent Settings.** We compare **CAML** with STGN with multi-agent settings, using ADR and EIR metrics across three accident-prone scenarios: (a) Overtaking, (b) Left Turn, and (c) Red Light Violation. *While STGN uses both RGB and depth data during testing, CAML relies solely on RGB, yet achieves comparable, or even better performance*. This highlights the effectiveness of **CAML**, leveraging LiDAR as an auxiliary modality during training to enhance performance.

**Modality-Efficient Superiority.** How does **CAML** compare with other approaches that have access to more modalities during testing? By modality-efficient superiority, we refer to a model's ability to achieve comparable or even superior performance using fewer modalities compared to other approaches that rely on a richer set of modalities. We evaluate **CAML** against STGN (Gao et al., 2024) with multi-agent settings. **CAML** uses only RGB data during testing but STGN uses both RGB and depth data. Both models are evaluated using the same metrics, ADR and EIR, across the three accident-prone scenarios. Despite the fact that STGN utilizes both RGB and depth data during testing, **CAML** achieves comparable, and in some cases superior, performance while relying solely on RGB data, as illustrated in Figure 4. Notably, **CAML** exceeds the ADR of STGN by 9.26% in the left-turn scenario, demonstrating that our model can enhance driving safety even when constrained to fewer modalities. This further underscores the strength of **CAML**, which effectively leverages LiDAR data as an auxiliary modality during training to boost performance, even when such data is unavailable during testing. The fact that **CAML** matches or exceeds the performance of a model that uses more data at test time highlights the efficacy of our multi-agent collaboration approach.

**System Generalizability.** How effectively does the system generalize when we have fewer agents during testing compared to training? (e.g., we have multi-agent collaboration during training but only single agent during testing). We test the case where multi-agent collaboration is used during training, but only a single agent is present during testing. This test is also motivated by practical constraints, where in many real-world situations, multi-vehicle connected systems are not available, we only have a single vehicle. But it is reasonable to have multi-vehicle connected systems with multiple modalities during training to develop robust models. After training, we can then apply the model on a single vehicle for inference or testing, which is very valuable in practice and provides a cost-effective solution.

We compare the performance with other baselines, using the same evaluation metrics of ADR and EIR, across three accident-prone scenarios. The comparison results are presented in Figure 5. As

shown, **CAML** with a single agent during testing outperforms the three baselines across all scenarios, for both ADR and EIR metrics. This demonstrates that even with single agent during testing, **CAML** remains highly effective, by utilizing the multi-agent collaboration and auxiliary modalities provided by the teacher model during training.

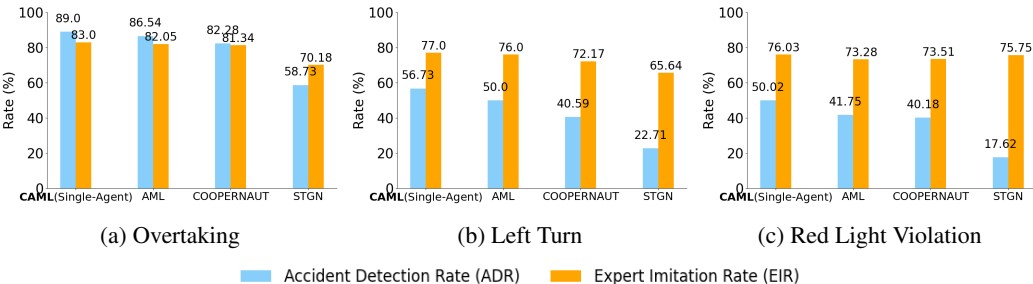

(a) Overtaking        (b) Left Turn        (c) Red Light Violation

Figure 5: **System Generalizability of CAML.** We evaluate the generalizability of **CAML** by testing the case where we have multi-agent collaboration during training, but only a single agent during testing. The performance is assessed using ADR and EIR across three accident-prone scenarios: (a) Overtaking, (b) Left Turn, and (c) Red Light Violation. *CAML with a single agent during testing consistently outperforms the three baselines across all scenarios, offering a valuable and cost-effective solution for practical applications.*

Overall, the experimental results clearly illustrate the superiority of our **CAML** framework. The ability of **CAML** to learn a more effective driving policy stems from the collaborative behavior of multiple agents, which together capture a wider and more nuanced representation of the data space. This broader coverage enables the ego vehicle to make better-informed decisions, improving safety and performance, particularly in complex, dynamic, and accident-prone environments where isolated agents with limited sensing capabilities might struggle.

## 5.2 Collaborative Semantic Segmentation for Aerial-Ground Vehicles

To further evaluate our approach, we focus on collaborative perception by conducting experiments with real-world data from aerial and ground vehicles for collaborative semantic segmentation. We use the dataset CoPeD (Zhou et al., 2024), with one aerial vehicle and one ground vehicle, in two different real-world scenarios of the indoor NYUARPL and the outdoor HOUSEA. For more details about the dataset, please refer to Zhou et al. (2024). Additionally, we introduce noise to the RGBD data collected by the ground vehicle. For both aerial and ground vehicles, RGB and depth data are used during training, while only RGB data is used during testing in **CAML**.

**Experimental Setup.** We adopt the FCN (Long et al., 2015) architecture as the backbone for semantic segmentation. We resize the input RGB and depth images to $224 \times 224$. To process RGB and depth data locally for each vehicle, we use ResNet-18 (He et al., 2016) as the encoder to extract feature maps of size $7 \times 7$. The RGB features from both vehicles are shared and fused through channel-wise concatenation, and the depth features are processed similarly. Then we apply $1 \times 1$ convolution to reduce the fused feature maps to the original channel dimensions for RGB and depth, respectively. We subsequently apply cross-attention to fuse the RGB and depth feature maps to generate multi-agent multi-modal feature aggregations. These aggregated features are passed through the decoder and upsampled to produce an output map matching the input image size.

We first train a teacher model offline for semantic segmentation with aerial-ground vehicles collaboration using cross-entropy loss, where each vehicle has both RGB and depth data. Then we train a student model to mimic the behavior of the teacher model with only RGB data for both aerial and ground vehicles through Knowledge Distillation (KD). The KD process is similar to that of the collaborative decision-making in CAV, please refer to Appendix A.4, but here we use a cross-entropy loss as the student task loss. For the detailed training settings, please see Appendix A.3.1.

We evaluate performance using the **Mean Intersection over Union (mIoU)** metric, which quantifies the average overlap between predicted segmentation outputs and ground truth across all classes. We compare the performance of **CAML** with AML (Shen et al., 2023) and FCN (Long et al., 2015). In the AML approach, only the ground vehicle operates, with RGB and depth data available during

training but only RGB data used for testing. The FCN approach involves only the ground vehicle operating with RGB data for both training and testing.

**Experimental Results.** We present the experimental results in Table 1. **CAML** demonstrates superior performance in terms of mIoU across both indoor and outdoor environments. Specifically, **CAML** achieves an improvement of mIoU for $7.4\%$ in indoor scenario and $10.8\%$ in outdoor scenario compared to AML (Shen et al., 2023). We also present the qualitative results in Fig. 7 in the Appendix. Despite the noisy input image from the ground vehicle, **CAML** produces predictions that are closest to the ground truth. This performance improvement can be attributed to **CAML**'s multi-agent collaboration, which provides complementary information to enhance data coverage and offers a more comprehensive understanding of the scenes. Additionally, the utilization of auxiliary depth data during training results in more precise segmentation outputs.

Table 1: Experimental results of semantic segmentation on real-world dataset CoPeD (Zhou et al., 2024) using aerial-ground vehicles in indoor and outdoor environments. **CAML** achieves the highest mIoU in both environments.

| Approach | mIoU (%) | |
|---|---|---|
| | Indoor | Outdoor |
| FCN (Long et al., 2015) | 51.20 | 56.22 |
| AML (Shen et al., 2023) | 55.89 | 60.32 |
| **CAML** | 60.05 | 66.83 |

## 6 CONCLUSIONS AND LIMITATIONS

In conclusion, we propose Collaborative Auxiliary Modality Learning (**CAML**), a novel framework that extends the Auxiliary Modality Learning (AML) paradigm to multi-agent systems. Unlike previous approaches that either concentrate on multi-agent collaboration without addressing modality reduction or handle multi-modality learning in single-agent settings, **CAML** unifies these concepts. It allows agents to collaborate in learning from shared modalities during training but enables efficient, modality-reduced inference. This approach not only reduces computational cost and data requirements at test time but also improves the accuracy of predictions by leveraging multi-agent collaboration. Furthermore, we systematically analyze the effectiveness of **CAML** from the perspective of uncertainty reduction and data coverage, providing a theoretical support to understand and explain why **CAML** works better than AML. We validate **CAML** on collaborative decision-making tasks for connected autonomous driving in complex and accident-prone scenarios, **CAML** achieves an improvement up to $58.3\%$ in accident detection compared to AML. Additionally, we validate our approach on real-world data from aerial-ground vehicles for collaborative semantic segmentation, achieving up to $10.8\%$ improvement in mIoU compared to AML. These significant enhancements in driving safety and semantic estimation underscore the practical implications of our framework.

Even though the advancements of **CAML**, there are some limitations and failure cases. One failure case is that if the modalities are misaligned, the model may struggle to perform effective fusion, leading to incorrect predictions. For instance, if modalities such as RGB and depth are captured at different time intervals or from non-overlapping fields of view, combining them effectively can be difficult. The auxiliary modalities or views from collaborative agents may become noise, useless or even degrading performance. Another limitation is the increasing system complexity. As the number of agents increases, the complexity of the system grows. The fusion of multi-agent and multi-modal data introduces challenges related to coordination overhead, which may lead to delays in the collaborative learning process.

Future work can focus on addressing these limitations and failure cases, and further exploring the applicability of **CAML** in other domains where multi-agent collaboration with multi-modal data is essential, and investigating additional strategies to enhance performance during testing with fewer modalities.

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

# A APPENDIX

## A.1 DATA COVERAGE

We present a qualitative example highlighting how multi-agent collaboration provides complementary information to enhance data coverage. In a red-light violation scenario for connected autonomous driving, as shown in the following figure, the ego vehicle's view is obstructed, rendering the occluded vehicle invisible. However, collaborative vehicles are able to detect the occluded vehicle, providing critical complementary information. This additional data helps the ego vehicle overcome its occluded view, enabling it to make more informed decisions and avoid potential collisions with the occluded vehicle.

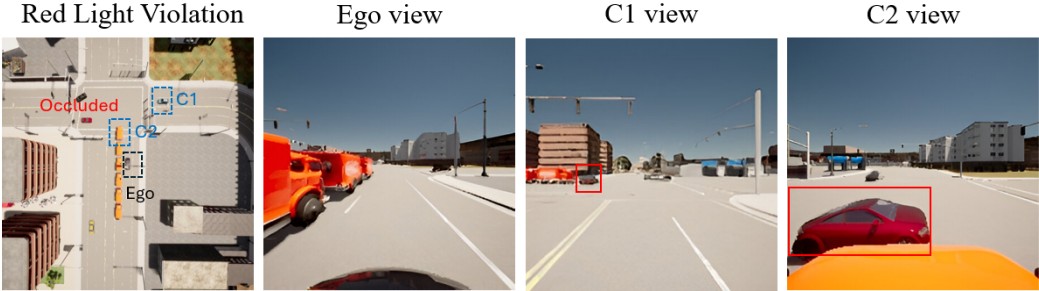

Figure 6: Qualitative example of multi-agent collaboration provides complementary information to enhance data coverage.

## A.2 REAL-WORLD AERIAL-GROUND SCENARIOS

### A.2.1 COLLABORATIVE SEMANTIC SEGMENTATION FOR AERIAL-GROUND VEHICLES

We present the qualitative results of collaborative semantic segmentation using real-world data from aerial-ground vehicles in the following figure. Despite the noisy input image from the ground vehicle, **CAML** produces predictions closest to the ground truth. This performance is attributed to its multi-agent collaboration, which provides complementary information to enhance viewpoints, and its utilization of multi-modal depth data during training, enabling more precise segmentation outputs.

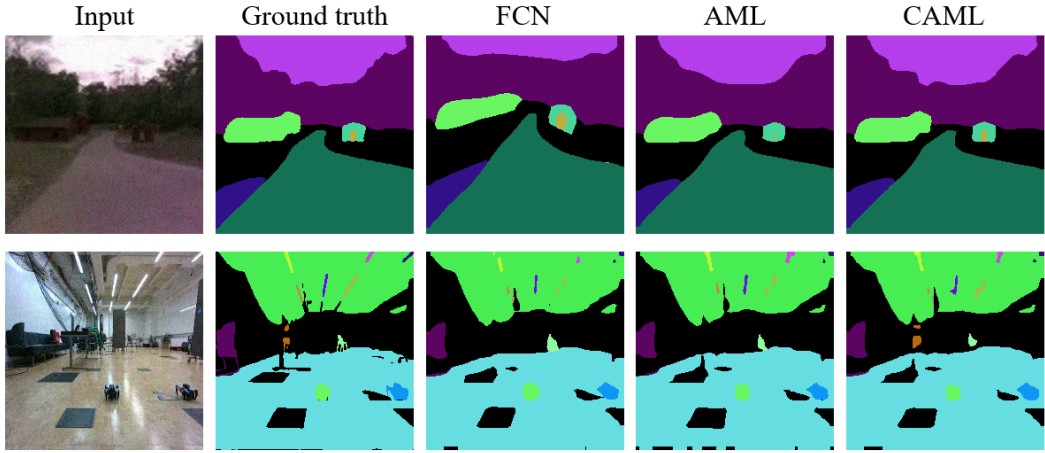

Figure 7: Qualitative results of different approaches on semantic segmentation on real-world data from aerial-ground vehicles in scenarios of both indoor and outdoor environments. From light to right, input image for the ground vehicle, ground truth segmentation map, FCN prediction, AML prediction, and **CAML** prediction. **CAML** prediction is the closest to the ground truth.

### A.2.2 ABLATION STUDIES

In the ablation studies, we explore another variant of **CAML** called Pre-fusion **CAML**, applied to the experiment of aerial-ground vehicles collaborative semantic segmentation. However, it is important to note that this variant can be applied to other domains and experiments as well. In this variant, each vehicle first locally extract feature maps of size $7 \times 7$ for both RGB and depth modalities. Instead of separately fusing the RGB and depth features between the vehicles, we first fuse the feature maps of RGB and depth within each single vehicle using cross-attention. Then we share and merge the fused RGBD features between vehicles via concatenation. We also apply $1 \times 1$ convolution to reduce the feature maps to the original channel dimensions. The multi-agent, multi-modal feature aggregations then pass through the decoder. Finally, we obtain the output map by upsampling to match the input image size. The mIoU of the Pre-fusion **CAML** is similar to that of **CAML**, achieving $59.16\%$ and $65.78\%$ for indoor and outdoor environments, respectively. By comparison, **CAML** achieves $60.05\%$ and $66.83\%$ in the same settings. Although the fusion order is different, both versions benefit from robust feature aggregation and multi-agent collaboration, which ultimately results in better segmentation performance.

Both **CAML** and its variant Pre-fusion **CAML** have their advantages, **CAML** fuses the same modalities across different agents, which provides better alignment because it ensures consistency in feature representation. And this approach is particularly beneficial when individual agent views are limited, as **CAML** effectively leverages diverse viewpoints to provide complementary information, enhancing overall data coverage. On the other hand, Pre-fusion **CAML** allows agent-specific contextual understanding by fusing different modalities locally within each agent. Furthermore, the system avoids redundant communication between agents by transmitting multi-modal aggregated features rather than modality-specific features separately. **CAML** can easily shift to Pre-fusion **CAML** because of the flexibility of our framework, depending on application scenarios.

### A.3 COMPLEXITY ANALYSIS

#### A.3.1 COMPARATIVE TRAINING COMPLEXITY

We report the training complexity of AML (Shen et al., 2023) and **CAML** for the experiments of collaborative decision-making in CAV and collaborative semantic segmentation for aerial-ground vehicles in Table 2 and Table 3, respectively. For the experiments, we employ a batch size of 32 and the Adam optimizer (Kingma, 2014) with an initial learning rate of $1e-3$, and a Cosine Annealing Scheduler (Loshchilov & Hutter, 2016) to adjust the learning rate over time. The model is trained on an Nvidia RTX 3090 GPU with AMD Ryzen 9 5900 CPU and 32 GB RAM for 200 epochs.

Table 2: Training complexity of AML and **CAML** in collaborative decision-making for connected autonomous driving.

| Approach | Parameters | Time/epoch |
|---|---|---|
| AML (Shen et al., 2023) | 19.5M | 34s |
| **CAML** | 39.3M | 73s |

Table 3: Training complexity of AML and **CAML** in collaborative semantic segmentation for aerial-ground vehicles.

| Approach | Parameters | Time/epoch |
|---|---|---|
| AML (Shen et al., 2023) | 13.5M | 3s |
| **CAML** | 25.5M | 7s |

#### A.3.2 TIME AND SPACE COMPLEXITY

In **CAML**, the agents' embeddings are shared based on whether the system operates in a centralized or decentralized manner. If the system is a centralized, all collaborative agents share their data with one designated ego agent for centralized processing. Each of the $N - 1$ collaborative agents performs its local computation independently, with a time complexity of $O(T_c)$ and a space

complexity of $O(S_c)$, where $T_c$ represents the time required for local computation, and $S_c$ is the associated space. Thus, the total computation time and space complexities for all collaborative agents are $O(T_c(N-1))$ and $O(S_c(N-1))$, respectively. For simplicity, assuming each communication from one collaborative agent to the ego agent consumes $O(D)$ time complexity and $O(M)$ space complexity, where $D$ is the time required for communication and $M$ is the corresponding space. Therefore, the total communication time and space complexities for gathering information at the ego agent are $O(D(N-1))$ and $O(M(N-1))$, respectively. Then the ego agent aggregates the received data, running a model, having a time and space complexity $O(T_e)$ and $O(S_e)$, where $T_e$ and $S_e$ represent the time and space required for the ego agent's computation. So the total time and space complexities are $O(T_c(N-1) + D(N-1) + T_e)$ and $O(S_c(N-1) + M(N-1) + S_e)$, respectively.

If the system is decentralized, each agent performs its local computation and shares information with other agents. For simplicity, let the local computation for a single agent has a time complexity of $O(T)$, where $T$ is the time required for local computation. Assume that communication from one agent to another agent requires $O(D)$ time complexity and $O(M)$ space complexity, where $D$ represents the time of communication between two agents, and $M$ denotes the space required for such communication. For $N$ agents, the total computation time complexity is $O(NT)$. In the worst case, each agent share data with all other agents, this can result in $O(N^2D)$ for pairwise sharing. So the total time complexity is $O(NT + N^2D)$. For space complexity, the storage requirement for all agents is $O(NS)$, where $S$ is the space needed per agent. Communication between agents adds an additional complexity of $O(N^2M)$. So the total space complexity is $O(NS + N^2M)$. In the typical case, if each agent communicates with only other $k$ agents ($k \ll N$) rather than all $N-1$ agents. The total time and space complexities become $O(NT + NkD)$ and $O(NS + NkM)$, respectively.

### A.4 KNOWLEDGE DISTILLATION

We begin by training a teacher decision-making model $\mathcal{T}$ offline using both RGB and LiDAR data, with a binary cross-entropy loss: $\mathcal{L}_{BCE}(y, \mathcal{T}) = -\mathbb{E}_{\mathcal{D}}\big[y_i \log(p_i) + (1-y_i)\log(1-p_i)\big]$, where $\mathcal{D}$ is the dataset, $y_i$ is the ground truth indicating whether the vehicle should brake, $p_i$ is the predicted probability by the teacher model $\mathcal{T}$. The student model $\mathcal{S}$ is trained to mimic the behavior of the teacher model while having less modalities. For each data point, the student model receives the same RGB image that the teacher model was given. The loss for the student model is a combination of two terms: the distillation loss using KL divergence between the student output and teacher output (soft targets), and the student task loss, which is the binary cross entropy loss between the student output and the true labels (hard targets). The soft targets from the teacher enrich learning with class similarities, while hard targets ensure alignment with true labels.

The soft targets are generated by applying a temperature scaling to the logits. The scaled logits are defined as: $z_i = \frac{\exp(z_i/t)}{\exp(z_0/t)+\exp(z_1/t)}$, where $z_i$ is the logit for class $i$ and $t$ is the temperature parameter. The distillation loss is defined as: $\mathcal{L}_{KD}(\mathcal{S}, \mathcal{T}) = -\sum z_i^{\mathcal{T}} \log(z_i^{\mathcal{S}})$, where $z_i^{\mathcal{T}}$ and $z_i^{\mathcal{S}}$ are the soft target probability from the teacher and student model, respectively. The overall loss for the student model is a weighted sum of the distillation loss and the binary cross-entropy loss: $\mathcal{L}_{\mathcal{S}} = (1-\alpha)\mathcal{L}_{BCE}(y, \mathcal{S}) + \alpha t^2 \mathcal{L}_{KD}(\mathcal{S}, \mathcal{T})$, where $\alpha$ is a hyperparameter that controls the trade-off between the two losses. We use $\alpha = 0.5$ and $t = 3.0$.

After the training of knowledge distillation process, we obtain a student model that uses only RGB data while learning from a teacher model that has access to both RGB and LiDAR data. This enables the student model to be effective during testing with only RGB data. Additionally, by leveraging knowledge distillation, the student model benefits from the additional insights provided by the LiDAR data during training, learning more effectively compared to training solely with RGB data.

