# OpenReview forum: "CAML: Collaborative Auxiliary Modality Learning for Multi-Agent Systems"
_ICLR.cc/2025/Conference — Submitted to ICLR 2025_

### Official Review · Reviewer_fWcH · 2024-10-21

**Soundness:** 3
**Presentation:** 3
**Contribution:** 3
**Rating:** 6
**Confidence:** 3

**Summary:**

This paper introduces Collaborative Auxiliary Modality Learning (CAML), a framework that extends Auxiliary Modality Learning (AML) to multi-agent systems. CAML allows agents to collaborate and share multimodal data during training, while enabling efficient, reduced-modality inference during testing. The authors provide theoretical analysis of CAML's effectiveness from the perspectives of uncertainty reduction and data coverage enhancement, explaining why CAML outperforms AML. The framework is validated through experiments on collaborative decision-making tasks for connected autonomous vehicles in accident-prone scenarios. The paper shows CAML's modality-efficient superiority and generalizability. The paper positions CAML as a unified approach that leverages multi-agent collaboration to improve robustness and accuracy in predictions, while reducing computational costs and data requirements at test time. This has potential applications in various domains where multi-agent collaboration and multimodal learning are crucial.

**Strengths:**

The paper demonstrates several notable strengths across the dimensions of originality, quality, clarity, and significance:



**Originality:** CAML creatively combines ideas from knowledge distillation, multi-agent systems, and multi-modal learning, resulting in a novel approach to handling complex, real-world scenarios.



**Quality:**The paper provides a solid theoretical analysis of CAML's effectiveness, examining it from the perspectives of uncertainty reduction and data coverage enhancement. This adds depth and credibility to the proposed approach.



**Clarity:** Well-structured: The paper follows a logical flow, clearly presenting the problem, methodology, theoretical analysis, and experimental results.



**Significance:** CAML's ability to perform well with reduced modalities during testing offers potential computational and resource efficiency benefits in deployed systems. While focused on autonomous driving, the framework's principles could be applied to other multi-agent, multi-modal learning scenarios, such as distributed sensing systems.



In summary, the paper presents an interesting approach to a problem in multi-agent systems, supported by thorough theoretical analysis and experimental validation.

**Weaknesses:**

Here are the key areas where the paper could be strengthened:

1. Modalities in Experimentation: It’s unclear which modalities are used for the experiments. While some sections mention depth, RGB, and LIDAR, the modalities should be clearly stated in Section 5.1 (Data Collection). Clarifying the exact modalities would provide better insight into the data used.

2. Overly Simplistic Driving Scenarios: The driving scenarios seem too basic. Including more complex environments with domain randomization (e.g., varying numbers of pedestrians and vehicles) would enrich the evaluation. Results from real-world scenarios would also be valuable to demonstrate generalization.

3. Limited Analysis of Failure Cases: The paper focuses on scenarios where CAML performs well, but lacks an exploration of its weaknesses or failure cases. To strengthen the paper, include an analysis of these cases and discuss possible improvements or limitations of the approach.

4. Insufficient Training Details: More details about the training setup would improve reproducibility. It would be useful to specify the GPU type, training time, and hardware resources used for training. Providing these details ensures transparency and supports future replication.

Typos:

1. L377: “CAML ." -> “CAML.”  There is a space between “L” and “.”

By addressing these points, the paper can offer a more thorough evaluation of CAML and broaden its contributions.

**Questions:**

Questions:

1. Clarification on Modalities: Can you clarify the exact modalities used in the experiments? Section 5.1 (Data Collection) does not specify this clearly, and it is inferred that depth, RGB, and LIDAR were utilized from other sections. Are these the only modalities involved, or are there others?

2. Complexity of Driving Scenarios: How realistic were the driving scenarios used in the experiments? Did you incorporate domain randomization, and how many pedestrians or vehicles were simulated? Additionally, are there plans to test the approach in real-world settings?

3. Failure Case Analysis: What are the specific failure cases where CAML underperforms? Including more analysis or examples of these cases would provide a clearer understanding of the framework’s limitations and potential areas for improvement.

4. Training Setup: Can you provide more details about the training setup, such as the GPU types, number of hours required for training, and hardware resources used? This would help improve reproducibility and enable more accurate comparisons in future work.

---

> ### Author Response · Authors · 2024-11-24
>
> Thanks for your detailed review and valuable feedback. We address your concerns below.
> > Weakness 1: Modalities in Experimentation: It’s unclear which modalities are used for the experiments. While some sections mention depth, RGB, and LIDAR, the modalities should be clearly stated in Section 5.1 (Data Collection). Clarifying the exact modalities would provide better insight into the data used.
>
> Thank you for your valuable feedback. In the experiments of collaborative decision-making in CAV, for each vehicle, RGB and LiDAR are used during training, while only RGB is used during testing. We have added exact modalities and clearly stated more experimental settings in the updated paper, please see general response **Question 2** and **Section 5.1 Data Collection**.
>
> > Weakness 2: Complexity of Driving Scenarios: How realistic were the driving scenarios used in the experiments? Did you incorporate domain randomization, and how many pedestrians or vehicles were simulated? Additionally, are there plans to test the approach in real-world settings?
>
> Thank you for your insightful feedback. The driving scenarios are indeed realistic which are built on AutoCast (Qiu et al., 2021, Cui et al., 2022) using CARLA, with a background traffic 30. They involve complex interactions such as overtaking, lane changing, and red-light violation, situations that are inherently accident-prone. And the number of connected vehicles is also varying, during the data collection, at each timestamp, the ego vehicle has a maximum of three collaborative vehicles provided their distance is within a threshold of 150 meter. We have also added new experiments with real-world data for aerial-ground vehicles collaborative semantic segmentation. Please see the general response **Question 1** and the updated **Section 5.2 and Appendix A.2**.
>
> > Weakness 3: Limited Analysis of Failure Cases: The paper focuses on scenarios where CAML performs well, but lacks an exploration of its weaknesses or failure cases. To strengthen the paper, include an analysis of these cases and discuss possible improvements or limitations of the approach.
>
> Thanks for your valuable comment. We have added a failure case analysis and limitations of the research in the updated paper, please see **Section 6 Conclusions and Limitations.**
>
> > Weakness 4: Insufficient Training Details: More details about the training setup would improve reproducibility. It would be useful to specify the GPU type, training time, and hardware resources used for training. Providing these details ensures transparency and supports future replication.
>
> Thanks for pointing this out. Please see training details in general response **Question 2** and we have added these in the updated paper.
>
> > Typos: L377: “CAML ." -> “CAML.” There is a space between “L” and “.”
>
> Thank you, this typo has been fixed.
>
> > Question 1: Clarification on Modalities: Can you clarify the exact modalities used in the experiments? Section 5.1 (Data Collection) does not specify this clearly, and it is inferred that depth, RGB, and LIDAR were utilized from other sections. Are these the only modalities involved, or are there others?
>
> We have added the clarification on modalities in the updated paper, please see general response **Question 2**.
>
> > Question 2: Complexity of Driving Scenarios: How realistic were the driving scenarios used in the experiments? Did you incorporate domain randomization, and how many pedestrians or vehicles were simulated? Additionally, are there plans to test the approach in real-world settings?
>
> The driving scenarios used in our experiments are indeed realistic, as they involve complex interactions such as overtaking, lane changing, and red-light violation, situations that are inherently accident-prone. These scenarios were designed to represent typical real-world conditions using AutoCast (Qiu et al., 2021, Cui et al., 2022) with a background traffic 30. Yes, we plan to test the approach in real-world settings in the future. And we have already added new experiments with real-world data of aerial-ground vehicles for collaborative semantic segmentation, please see the general response **Question 1** and paper **Section 5.2 and Appendix A.2**.
>
> > Question 3: Failure Case Analysis: What are the specific failure cases where CAML underperforms? Including more analysis or examples of these cases would provide a clearer understanding of the framework’s limitations and potential areas for improvement.
>
> We have added failure case analysis and limitations, please see general response **Question 3 and Section 6 Conclusions and Limitations** in the updated paper.

---

> ### Author Response · Authors · 2024-11-24
> **Continuing**
>
> > Question 4: Training Setup: Can you provide more details about the training setup, such as the GPU types, number of hours required for training, and hardware resources used? This would help improve reproducibility and enable more accurate comparisons in future work.
>
> Please refer to general response **Question 2** and we have added the training details in the updated **Appendix A.3.1**.
>
> [1] Hang Qiu, Pohan Huang, Namo Asavisanu, Xiaochen Liu, Konstantinos Psounis, and Ramesh Govindan. Autocast: Scalable infrastructure-less cooperative perception for distributed collaborative driving. arXiv preprint arXiv:2112.14947, 2021.
>
> [2] Jiaxun Cui, Hang Qiu, Dian Chen, Peter Stone, and Yuke Zhu. Coopernaut: End-to-end driving with cooperative perception for networked vehicles. In Proceedings of the IEEE/CVF Conference on Computer Vision and Pattern Recognition, pp. 17252–17262, 2022.

---

> > ### Comment · Reviewer_fWcH · 2024-11-26
> >
> > Thank you for your response. The author's explanation has addressed my concern.

---

> > > ### Author Response · Authors · 2024-11-26
> > >
> > > Dear Reviewer,
> > >
> > > Thank you for the kind feedback. We appreciate your time and expertise in reviewing our work.
> > >
> > > Best regards,
> > > Authors

---

> > > > ### Author Response · Authors · 2024-11-28
> > > >
> > > > Dear Reviewer,
> > > >
> > > > Since we have addressed your concern and also revised the paper based on your constructive feedback, could you please consider updating your score accordingly? We are deeply grateful for that.
> > > >
> > > > Thank you again for taking the time to review our work, your insights have been incredibly helpful to our research.
> > > >
> > > > Best,
> > > > Authors

---

> ### Author Response · Authors · 2024-12-01
>
> Dear Reviewer,
>
> As the rebuttal is going to end soon, could you please kindly let us know whether our response and the revised paper with additional detailed experiments and clarification have influenced your evaluation?
>
> Thank you for taking the time to review our work, your insights have been incredibly helpful to our research. And we would be deeply grateful if you could update the rating since we addressed your concern.
>
> Sincerely,
> The Authors

---

### Official Review · Reviewer_97B9 · 2024-11-04

**Soundness:** 3
**Presentation:** 2
**Contribution:** 2
**Rating:** 5
**Confidence:** 3

**Summary:**

This paper attempts to address the Auxiliary Modality Learning problem by means of a multi-agent system, where models are permitted to employ additional modalities during training while enabling inference with fewer or even a single modality.

**Strengths:**

1. The topic of  Auxiliary Modality Learning is interesting and practical. It provides solutions to deal with scenarios where limited modalities are available for inference, making it applicable in real-world situations with constrained data.

2. The author's attempt to address AML with a multi-agent system is interesting. The use of a multi-agent system in this context is an innovative concept that can lead to more efficient and effective learning processes.

3. The analysis part puts forward three interesting questions. These questions cover important aspects of agent collaboration and can lead to a deeper understanding and evaluation of the related mechanisms and effects.

**Weaknesses:**

1. Many experimental settings are not described clearly. For instance, is there a correlation between the number of agents and the number of modalities? The number of agents appears not to be explicitly stated in the experiment. Additionally, I am curious about how the experiment is devised when the number of modalities exceeds 3, and what the outcome would be.

2. Accurate numbers should preferably be marked on the histograms in the experiment for convenient viewing. For example, in Figure 4(a), it is difficult to tell the highs and lows. Meanwhile, I'm also curious about why while STGN uses both RGB and depth data during testing, CAML relies solely on RGB, yet achieves comparable, or even better performance.

3. There should be a comparative analysis of training complexity. Multiple agents inevitably increase the parameters of the network. Is the source of the performance improvement due to the increase in network parameters?

4. Discussions on limitations are necessary as there is no description of the limitations of the proposed method.

**Questions:**

Please see weaknesses above.

---

> ### Author Response · Authors · 2024-11-24
>
> Thanks for your review and valuable feedback. We address the concerns below.
>
> > Weakness 1: Many experimental settings are not described clearly. For instance, is there a correlation between the number of agents and the number of modalities? The number of agents appears not to be explicitly stated in the experiment. Additionally, I am curious about how the experiment is devised when the number of modalities exceeds 3, and what the outcome would be.
>
> We appreciate your attention to the clarity of the experimental settings. To clarify, the number of agents does not correlate with the number of modalities. The modalities are independently handled by each agent locally with respective encoders (e.g., RGB, LiDAR). We have clarified this in the updated paper **Section 3** and we also explicitly stated the number of agents in the experiments in **Section 5.1 Data Collection**.
>
> As for handling more than three modalities, we believe that the CAML framework can still work, such as text, audio, vision and other sensor inputs, without fundamental limitations. We plan to explore the integration of more diverse modalities as a direction for future work.
>
> > Weakness 2: Accurate numbers should preferably be marked on the histograms in the experiment for convenient viewing. For example, in Figure 4(a), it is difficult to tell the highs and lows. Meanwhile, I'm also curious about why while STGN uses both RGB and depth data during testing, CAML relies solely on RGB, yet achieves comparable, or even better performance.
>
> Thank you for your valuable feedback. Accurate numbers have been marked on the histograms in the updated Figure 3, 4 and 5. Regarding your question that CAML achieves comparable performance with STGN, we believe there are two main reasons. First, CAML effectively leverages LiDAR data as an auxiliary modality during training to boost performance through knowledge distillation. Second, in STGN, depth is used to predict object 3D bounding boxes with YOLO, and the predicted 3D position is then used for downstream tasks. But the YOLO model is only pretrained, not finetuned, so it may generate less accurate 3D positions.
>
> > Weakness 3: There should be a comparative analysis of training complexity. Multiple agents inevitably increase the parameters of the network. Is the source of the performance improvement due to the increase in network parameters?
>
> Thank you for the insightful comment. We have added the comparative training complexity analysis of AML and CAML in the updated paper, please see **Appendix A.3.1**. It is true that incorporating multiple agents introduces additional parameters to the system, the primary source of performance improvement in CAML is not solely due to the increase in network complexity. Instead, the performance gains come primarily from the collaborative nature of the multi-agent collaboration, where agents share and aggregate information across modalities, thus improving the overall robustness and accuracy of predictions.
>
> > Weakness 4: Discussions on limitations are necessary as there is no description of the limitations of the proposed method.
>
> Thank you for the valuable comment. We have added discussions on limitations in the paper, please see general response **Question 3 and Section 6 Conclusions and Limitations**.

---

> > ### Author Response · Authors · 2024-11-26
> >
> > Dear Reviewer,
> >
> > Thanks for your constructive review and valuable comments.
> >
> > If you have further questions or comments, please do not hesitate to let us know, as tomorrow is the deadline for uploading a revised PDF.
> >
> > We hope that our responses sufficiently address your concerns. If so, we would be really grateful if you could consider adjusting your score accordingly.
> >
> > We greatly appreciate your time and effort in reviewing our work and look forward to any further feedback.
> >
> > Best,
> > Authors

---

> > > ### Author Response · Authors · 2024-11-27
> > >
> > > Dear Reviewer,
> > >
> > > Apart from the response, we have also updated the paper based on your constructive feedback. Today is the last day to submit a revised PDF.
> > >
> > > We hope that our responses and revisions have addressed your concerns. We really appreciate if you could consider adjusting your score accordingly.
> > >
> > > We deeply appreciate your time and effort in reviewing our work and look forward to any further feedback you may have.
> > >
> > > Best regards,
> > > The Authors

---

> > > > ### Author Response · Authors · 2024-12-01
> > > >
> > > > Dear Reviewer,
> > > >
> > > > As the rebuttal is going to end soon, could you please confirm if you’ve reviewed our response and the revised paper. Please kindly let us know whether our response and the revised paper with additional detailed experiments and clarification have influenced your evaluation?
> > > >
> > > > Thank you for taking the time to review our work, your insights have been incredibly helpful to our research. And we would be deeply grateful if you could update the rating if we addressed your concern.
> > > >
> > > > Sincerely,
> > > > The Authors

---

### Official Review · Reviewer_8JqU · 2024-11-04

**Soundness:** 3
**Presentation:** 3
**Contribution:** 2
**Rating:** 5
**Confidence:** 4

**Summary:**

The paper aims to generalize Auxiliary Modality Learning (AML) to the context of cooperative perception. AML benefits machine learning by enabling inferencing on reduced modality space. The paper validates its proposed CAML method from the perspective of effective uncertainty reduction and data coverage to justify its superiority over AML. Empirical experiments on accident detection further support the conclusion.

**Strengths:**

The paper is well-organized, with a clear objective and statement of its design logic. The experiment design comparing the proposed CAML with its baseline AML effectively showcased the benefits of extending to a cooperative context well.

**Weaknesses:**

1. This paper's main contribution is an extension of an existing maturely developed machine learning technique, AML, to the context of cooperative perception, which limits its scope of novelty. Despite experiments conducted by the authors to investigate the benefits of CAML compared to single-agent AML, they only investigate one of the plausible paradigms for applying the AML - ego-only paradigm. Since we are looking into a cooperative perception task, will it be more sensical to use AML for training the encoder for single vehicles in parallel before fusion (**pre-fusion**) and compare this application with the proposed CAML, which is essentially applying a joint-version of AML to train the fusion module (**on-fusion**)?
2. The authors have cited several related works to demonstrate the benefits of applying AML to a perception system. However, it would be much better if the authors could present a comparative study of CAML, AML, and other cooperative perception methods on traditional cooperative perception tasks, such as objective detection, to showcase the need for applying AML in cooperative perception.

**Questions:**

Problems have been raised in the discussion about the weakness of this paper. Here are a few of them in short:
1. Can you justify the need to use AML for cooperative perception in general? If it was beneficial, why had no one before this paper explored using it for more traditional tasks such as cooperative object detection?
2. What is CAML's time and space complexity concerning the number of connected agents? From your description, I presume it is quadratic. Would this trade-off in extra time and space complexity be worth it?
3. Can you compare applying AML on training the entire fusion module to using it only for the encoder and decoder on single vehicles (pre- or post-fusion)?

---

> ### Author Response · Authors · 2024-11-24
>
> Thanks for your review and we address your concerns below.
> > Weakness 1: This paper's main contribution is an extension of an existing maturely developed machine learning technique, AML, to the context of cooperative perception, which limits its scope of novelty. Despite experiments conducted by the authors to investigate the benefits of CAML compared to single-agent AML, they only investigate one of the plausible paradigms for applying the AML - ego-only paradigm. Since we are looking into a cooperative perception task, will it be more sensical to use AML for training the encoder for single vehicles in parallel before fusion (pre-fusion) and compare this application with the proposed CAML, which is essentially applying a joint-version of AML to train the fusion module (on-fusion)?
>
> We appreciate your feedback and the suggestion regarding different paradigms for applying AML.
>
> First, we would like to emphasize that our proposed Collaborative Auxiliary Modality Learning (CAML) is not merely an incremental extension of Auxiliary Modality Learning (AML). CAML specifically addresses the unique challenges of multi-agent systems, including heterogeneity in sensing modalities, dynamic team compositions, and team-level decision-making with uncertainty reduction. CAML has broader applications of connected autonomous driving, collaborative search and rescue, as well as intelligent transportation. To the best of our knowledge, this work is the first to propose a flexible and principled framework capable of tackling a broad range of collaborative tasks in multi-agent systems where limited modalities are available for inference, especially in real-world scenarios.
>
> Second, to evaluate different paradigms for applying AML, we have further conducted new experiments involving real-world data from aerial-ground vehicles for collaborative semantic segmentation.  We added a variant of CAML called Pre-fusion CAML as ablation studies where we apply AML on single-agent to fuse different modalities first. The mIoU of the Pre-fusion CAML is similar to that of CAML. Although the fusion order is different, both versions benefit from robust feature aggregation and multi-agent collaboration, which ultimately results in better segmentation performance. Please see the general response **Question 1** for more details, we also updated in the paper **Section 5.2 and Appendix A.2**.
>
> > Weakness 2: The authors have cited several related works to demonstrate the benefits of applying AML to a perception system. However, it would be much better if the authors could present a comparative study of CAML, AML, and other cooperative perception methods on traditional cooperative perception tasks, such as objective detection, to showcase the need for applying AML in cooperative perception.
>
> We appreciate the reviewer's suggestion. We have conducted new experiments involving real-world data from aerial-ground vehicles for collaborative semantic segmentation. The real-world dataset does not just cover the traditional cooperative perception tasks, but also cover the aerial-ground scenarios, which is more inspiring to see the benefits of CAML. Please see the general response **Question 1** for more details, we also updated in the paper **Section 5.2 and Appendix A.2**.
>
> > Question 1: Can you justify the need to use AML for cooperative perception in general? If it was beneficial, why had no one before this paper explored using it for more traditional tasks such as cooperative object detection?
>
> The use of AML is particularly valuable in scenarios with constrained data where limited modalities are available for inference. Especially in real world, we cannot always guarantee that we have the same data modalities in both training and testing, some data modalities may be missing during testing compared to training. AML addresses this issue by improving the system's robustness to missing or incomplete modalities during inference. Second, AML promotes computation and resource efficiency during testing, which is very important for deployment in real world. Also as pointed out by Reviewer fWcH, the significance of CAML's principles can be applied to other multi-agent, multi-modal learning scenarios.
>
> There are actually some previous work of exploring AML or similar ideas to perception tasks (Shen et al., 2023; Hoffman et al., 2016; Wang et al., 2018; Garcia et al., 2018, 2019; Piasco et al., 2021), as we described in the Related Work. AML is still a relatively new research area. Shen et al. (2023) formalized various related research problems as the AML framework. But these earlier works have primarily focused on single-agent settings, which is not collaborative. In this paper, we took a novel effort to integrate multi-agent systems into AML, which can help deal with more complex real-world scenarios.

---

> ### Author Response · Authors · 2024-11-24
> **Continuing**
>
> > Question 2: What is CAML's time and space complexity concerning the number of connected agents? From your description, I presume it is quadratic. Would this trade-off in extra time and space complexity be worth it?
>
> In CAML, the agents' embeddings are shared based on whether the system operates in a centralized or decentralized manner. If the system is a centralized, all collaborative agents share their data with one designated ego agent for centralized processing. Each of the $N-1$ collaborative agents performs its local computation independently, contributing $O(T_c)$ time and $O(S_c)$ space each. Thus, the total computation time and space for all collaborative agents are $O(T_c(N-1))$ and $O(S_c(N-1))$, respectively. For simplicity, assuming each communication from one collaborative agent to the ego agent consumes $O(D)$ time and $O(M)$ space, the total communication time and space complexity for gathering information at the ego agent are $O(D(N-1))$ and $O(M(N-1))$, respectively. Then the ego agent aggregates the received data, running a model, having a time and space complexity $O(T_e)$ and $O(S_e)$, respectively. So the total time and space complexity are $O(T_c(N-1)+D(N-1)+T_e)$ and $O(S_c(N-1)+M(N-1)+S_e)$, respectively.
>
> If the system is decentralized, each agent performs its local computation and shares information with other agents. For simplicity, let the computation time for each agent be $O(T)$, and assume that communication from one agent to another agent requires $O(D)$ time and $O(M)$ space. For $N$ agents, the total computation time is $O(NT)$. In the worst case, each agent share data with all other agents, this can result in $O(N^2D)$ for pairwise sharing. So the total time complexity is $O(NT+N^2D)$. For space complexity,  the storage requirement for all agents is $O(NS)$, where $S$ is the storage per agent. Communication between agents adds an additional complexity of $O(N^2M)$. So the total space complexity is $O(NS+N^2M)$. In the typical case, if each agent communicates with only other $k$ agents ($k \ll N$) rather than all $N-1$ agents. The total time and space complexity becomes $O(NT+NkD)$ and $O(NS+NkM)$, respectively. We have updated these in the paper, please see **Appendix A.3.2.**
>
> The trade-off in extra time and space complexity is worth it. By having multi-agent collaboration, we can capture a more comprehensive understanding of the environment and have improved robustness and more accurate prediction. These are especially important for real-world scenarios where performance is critical like autonomous driving and collaborative perception, otherwise may cause accident.
>
> > Question3: Can you compare applying AML on training the entire fusion module to using it only for the encoder and decoder on single vehicles (pre- or post-fusion)?
>
> Thanks for your valuable suggestion. We have conducted new experiments involving real-world data from aerial-ground vehicles for collaborative semantic segmentation. And we added pre-fusion version of CAML as ablation studies and compared with the proposed CAML. Please see the general response **Question 1** for more details, we also updated in the paper **Section 5.2 and Appendix A.2**.
>
> [1] Judy Hoffman, Saurabh Gupta, and Trevor Darrell. Learning with side information through modality hallucination. In Proceedings of the IEEE conference on computer vision and pattern recognition, pp. 826–834, 2016.
>
> [2] Lan Wang, Chenqiang Gao, Luyu Yang, Yue Zhao, Wangmeng Zuo, and Deyu Meng. Pm-gans: Discriminative representation learning for action recognition using partial-modalities. In Proceedings of the European Conference on Computer Vision (ECCV), pp. 384–401, 2018.
>
> [3] Nuno C Garcia, Pietro Morerio, and Vittorio Murino. Modality distillation with multiple stream networks for action recognition. In Proceedings of the European Conference on Computer Vision (ECCV), pp. 103–118, 2018.
>
> [4] Nuno C Garcia, Pietro Morerio, and Vittorio Murino. Learning with privileged information via adversarial discriminative modality distillation. IEEE transactions on pattern analysis and machine intelligence, 42(10):2581–2593, 2019.
>
> [5] Nathan Piasco, Desire Sidibe, Valerie Gouet-Brunet, and Cedric Demonceaux. Improving image description with auxiliary modality for visual localization in challenging conditions. International Journal of Computer Vision, 129(1):185–202, 2021.

---

> > ### Comment · Reviewer_8JqU · 2024-11-26
> >
> > Thank you for your efforts to address my questions. I find it a bit tricky to follow your big-O notations at first glance in the revised version. It would be nice if you could specify definitions of symbols such as $T_{c}$, $S_{c}$, and $S_{e}$ to enable the general audience to understand the context here.
> > Meanwhile, I do not entirely agree with your assertion that "...we can capture a more comprehensive understanding of the environment and have improved robustness..." Collaboration does not necessarily guarantee improved robustness. A related area is "Network Vulnerability Analysis." I did not see a design in your experiment that specifically investigated the system's sensitivity toward single-point or multi-point failures in a collaborative network. Therefore, it is irresponsible to make the assertion directly.

---

> ### Author Response · Authors · 2024-11-26
>
> Dear Reviewer,
>
> Thanks for your valuable comment. We have specified definitions of all symbols in the updated paper **Appendix A.3.2** and also removed the assertion about robustness on Line 774. If you have further questions or comments, please do not hesitate to let us know, as tomorrow is the deadline for uploading a revised PDF.
>
> If you feel that our response and revisions adequately address your concerns, we would be really grateful if you could consider adjusting your score accordingly.
>
> We greatly appreciate your time and effort in reviewing our work and look forward to any further feedback.
>
> Best,
> Authors

---

> > ### Author Response · Authors · 2024-11-27
> >
> > Dear Reviewer,
> >
> > Apart from the response, we have also updated the paper based on your constructive feedback. Today is the last day to submit a revised PDF.
> >
> > We hope that our responses and revisions have addressed your concerns. We really appreciate if you could consider adjusting your score accordingly.
> >
> > We deeply appreciate your time and effort in reviewing our work and look forward to any further feedback you may have.
> >
> > Best regards,
> > The Authors

---

> > > ### Author Response · Authors · 2024-12-01
> > >
> > > Dear Reviewer,
> > >
> > > As the rebuttal is going to end soon, could you please kindly let us know whether our response and the revised paper with additional detailed experiments and clarification have influenced your evaluation?
> > >
> > > Thank you for taking the time to review our work, your insights have been incredibly helpful to our research. And we would be deeply grateful if you could update the rating since we addressed your questions.
> > >
> > > Sincerely,
> > > The Authors

---

### Official Review · Reviewer_bryx · 2024-11-04

**Soundness:** 2
**Presentation:** 3
**Contribution:** 2
**Rating:** 5
**Confidence:** 3

**Summary:**

This paper extends Auxiliary Modality Learning (AML) for multi-agent systems, resulting in Collaborative Auxiliary Modality Learning (CAML). Then each single-agent can maintain robustness robustness even with missing modalities during inference through multi-agent information sharing. Analysis about uncertainty reduction, and data coverage provides a theoretical support to understand and explain why CAML works better than AML. And the CAML system is validated in the application of decision-making for CAV in accident-prone scenarios, demonstrating the effectiveness of the proposed CAML.

**Strengths:**

1. The writing is easy to follow.
2. The motivation and proposed method are clearly stated.
3. This work represents a novel effort to integrate multi-agent systems into Auxiliary Modality Learning.
4. The experiments demonstrate the effectiveness of CAML compared to previous baselines by incorporating multi-agent collaboration.
5. It is promising that CAML can achieve comparable performance with RGBD using only RGB modality, and the benefits of multi-agent training can extend to single-agent testing scenarios.

**Weaknesses:**

1. Lack of ablation studies on a wider range of modalities. For instance, what happens when training involves both LiDAR and RGB modalities, while testing uses only RGB or LiDAR? Would the performance still improve when the modality gap is significant?
2. Lack of validation on real-world benchmarks. Validating real-world data is crucial, especially for applications in connected autonomous vehicles. Numerous real-world benchmarks exist, such as DAIR-V2X, V2V4Real, and TUMTraf, making real-world validation essential.
3. Lack of quantitative analysis regarding uncertainty reduction and data coverage. Additionally, providing qualitative visualization examples would enhance the presentation.

**Questions:**

1. If different modality embeddings are combined through concatenation, can the system only accommodate a fixed modality?
2. What about heterogeneous scenarios where agents possess different modalities, such as in an RGBD context, including single agents with either RGB or depth data?

---

> ### Author Response · Authors · 2024-11-24
>
> Thanks for your review and we really appreciate your comments. We address your concerns below.
> > Weakness 1: Lack of ablation studies on a wider range of modalities. For instance, what happens when training involves both LiDAR and RGB modalities, while testing uses only RGB or LiDAR? Would the performance still improve when the modality gap is significant?
>
> We appreciate the reviewer's insightful comment. We would like to point out that we already have the case of training with both LiDAR and RGB modalities, while testing using only RGB, in the experiments of collaborative decision-making in CAV, for both CAML and AML ( Please see **Section 5.1 Baseline Comparison in the paper**).
>
>  We have conducted more experiments where we have both LiDAR and RGB for both training and testing, which we refer to as the teacher models for both **single-agent** and **multi-agent** settings, as shown below. Then we compare the performance of ADR (accident detection rate) and EIR (expert imitation rate) with AML and CAML. The performances of both CAML and AML decrease compared to respective teacher models because the missing LiDAR modality during testing, but the performance still improves by up to **58.3%** when comparing CAML to AML. This addresses the concern in modality gaps, while the performance still improves when comparing AML vs. CAML.
>
> | Approach | Overtaking | | Left Turn | | Red Light Violation | |
> |-----------|----------------|------------:|-------------|---------:|----------|------------:|
> |           | ADR↑ | IR↑ | ADR↑ | IR↑ | ADR↑ | IR↑ |
> | Single-Agent | | | | | | |
> | Teacher (Training: RGB + LiDAR, Testing: RGB + LiDAR) | 0.8654 | 0.8205 | 0.5576 | 0.7483 | 0.5250 | 0.7504 |
> | AML (Training: RGB + LiDAR, Testing: RGB) | 0.8206 | 0.8322 | 0.5000 | 0.7600 | **0.4175** | 0.7328 |
> | Multi-Agent | | | | | | |
> | Teacher (Training: RGB + LiDAR, Testing: RGB + LiDAR) | 0.9604 | 0.8676 | 0.7329 | 0.8122 | 0.6875 | 0.8578 |
> | CAML (Training: RGB + LiDAR, Testing: RGB) | 0.9288 | 0.8381 | 0.6632 | 0.7946 | **0.6607** | 0.8262 |
>
> We also validate with new experiments and real-world data, with more modalities such as RGBD and ablation studies, the experiments of the collaborative semantic segmentation for aerial-ground vehicles. Please see the general response **Question 1** for more details, we also updated in the paper **Section 5.2 and Appendix A.2**.
>
> > Weakness 2: Lack of validation on real-world benchmarks. Validating real-world data is crucial, especially for applications in connected autonomous vehicles. Numerous real-world benchmarks exist, such as DAIR-V2X, V2V4Real, and TUMTraf, making real-world validation essential.
>
> We appreciate the reviewer’s feedback highlighting the importance of validating our approach on real-world data. To address this, we have conducted new experiments involving real-world data from aerial-ground vehicles for collaborative semantic segmentation. Please see the general response **Question 1** for more details, we also updated in the paper **Section 5.2 and Appendix A.2** for more results and ablation studies.
>
> > Weakness 3: Lack of quantitative analysis regarding uncertainty reduction and data coverage. Additionally, providing qualitative visualization examples would enhance the presentation.
>
> We appreciate the reviewer’s insights regarding the need for quantitative analysis of uncertainty reduction and data coverage, as well as qualitative visualization. We would like to point out that our experimental results do provide quantitative evidence of these aspects, as demonstrated by the improved prediction accuracy, which reflects reduced uncertainty and enhanced data coverage facilitated by multi-agent collaboration. Additionally, we added qualitative visualization examples in the paper to enhance the presentation, please see **Appendix A.1 Data Coverage**, where we show in a red-light violation scenario in connected autonomous driving, the ego vehicle’s view is obstructed, rendering the occluded vehicle invisible. However, collaborative vehicles are able to detect the occluded vehicle, providing critical complementary information. This additional data helps the ego vehicle overcome its occluded view, enabling it to make more informed decisions and avoid potential collisions with the occluded vehicle.
>
> > Question 1: If different modality embeddings are combined through concatenation, can the system only accommodate a fixed modality?
>
> Thank you for your question. We would like to emphasize that CAML is a general framework and it can accommodate various modalities. Concatenation does not restrict the framework to a fixed modality, any modality embeddings can be concatenated as long as the embeddings are compatible in terms of their dimensionality after extraction from the respective encoders. And different embeddings can also be combined in other ways like cross-attention, concatenation is one of them.

---

> ### Author Response · Authors · 2024-11-24
> **Continuing**
>
> > Question 2: What about heterogeneous scenarios where agents possess different modalities, such as in an RGBD context, including single agents with either RGB or depth data?
>
> Thank you for raising this point. CAML can handle heterogeneous scenarios where agents possess different modalities. As we described and explained in Section 3 of the paper, each agent can process a different number of modalities during training, different agents can also have different main modalities and auxiliary modalities.

---

> > ### Author Response · Authors · 2024-11-26
> >
> > Dear Reviewer,
> >
> > Thank you for your valuable review and feedback.
> >
> > If you have any additional questions or comments, please feel free to let us know, as the deadline for submitting a revised PDF is tomorrow.
> >
> > We hope that our responses have addressed your concerns. If so, we would be sincerely grateful if you could consider adjusting your score accordingly.
> >
> > We deeply appreciate your time and effort in reviewing our work and look forward to any further feedback you may have.
> >
> > Best regards,
> > The Authors

---

> ### Author Response · Authors · 2024-11-27
>
> Dear Reviewer,
>
> Apart from the response, we have also updated the paper based on your constructive feedback. Today is the last day to submit a revised PDF.
>
> We hope that our responses and revisions have addressed your concerns. We really appreciate if you could consider adjusting your score accordingly.
>
> We deeply appreciate your time and effort in reviewing our work and look forward to any further feedback you may have.
>
> Best regards,
> The Authors

---

> > ### Comment · Reviewer_bryx · 2024-11-29
> > **Response to Authors by Reviewer bryx**
> >
> > Thank you for the detailed experiments and explanations. Most of my concerns have been addressed. However, I have one main follow-up regarding the real-world evaluation.
> >
> > 1. Regarding experiments on the real-world collaborative perception benchmark, CoPed is a promising new benchmark. However, it lacks comparative baselines. Conducting experiments on more established collaborative perception benchmarks, such as DAIR-V2X, V2V4Real, and TUMTraf, which include more baseline comparisons, would strengthen the findings and make the results more convincing.

---

> ### Author Response · Authors · 2024-11-29
>
> Dear Reviewer,
>
> The reason we chose the CoPed dataset for real-world collaborative perception is that it covers different scenarios, the aerial-ground multi-robot collaboration scenarios, which is most suited for collaborative, multi-agent decision-making when using CAML. We also have experiments of autonomous ground vehicles with different sensors, another common practical scenario, so we use different scenarios to validate our approach. We will validate on benchmarks like DAIR-V2X, V2V4Real, and TUMTraf in future work.
>
> Since we have addressed most of your concerns with detailed experiments and explanations, could you please consider updating the score accordingly? We are deeply grateful for that.

---

> > ### Author Response · Authors · 2024-12-01
> >
> > Dear Reviewer,
> >
> > As the rebuttal is going to end soon, could you please kindly let us know whether our response and the revised paper with additional detailed experiments and clarification have influenced your evaluation?
> >
> > Thank you for taking the time to review our work, your insights have been incredibly helpful to our research. And we would be deeply grateful if you could update the rating since we addressed most of your concerns.
> >
> > Sincerely,
> > The Authors

---

### Author Response · Authors · 2024-11-24
**General Response**

Dear Area Chairs and Reviewers,

We thank all the reviewers for their detailed and helpful comments on our work. We appreciate the reviewers for acknowledging our strengths and contributions, such as
- a **novel and interesting approach** to integrate multi-agent systems into Auxiliary Modality Learning (bryx, 97B9, fWcH);
- **practical solutions** (97B9) to deal with real-world scenarios where limited modalities are available for inference;
- **significance** (fWcH) of the proposed approach being applied to other multi-agent, multi-modal learning scenarios;
- **interesting and solid theoretical analysis** which leads to a deeper understanding and adds depth and credibility to the proposed approach (97B9, fWcH);
- **clear motivation, method and logic** (bryx, 8JqU, fWcH);
- experiments demonstrating the **effectiveness** of CAML (bryx, 8JqU); and
- **well written and organized** (bryx, 8JqU, fWcH).

Below we address the concerns raised by the Reviewers.  We include more requested experiments with real-world data and more ablation studies.  We also provide more discussions on limitations and failure case analysis, more experimental and training details, and add clarity and depth to address any ambiguities. Our responses to specific concerns are detailed below. We thank you all for the opportunity to improve our work with your constructive feedback.

Best regards,
The Authors
___
Here we refer to some general questions:

**Question 1 (bryx, 8JqU, fWcH): Experiments on real-world data and other collaborative perception tasks.**

**Experimental Setup.** We conducted additional experiments with real-world data from aerial-ground vehicles for collaborative semantic segmentation. We use the dataset CoPeD (Zhou et al., 2024), with one aerial vehicle and one ground vehicle, in two different real-world scenarios of the indoor NYUARPL and the outdoor HOUSEA. For more details about the dataset, please refer to Zhou et al. (2024). Additionally, we introduce noise to the RGBD data collected by the ground vehicle. For both aerial and ground vehicles, RGB and depth data are used during training, while only RGB data is used during testing in CAML.

We adopt the FCN (Long et al., 2015) architecture as the backbone for semantic segmentation. We resize the input RGB and depth images to $224\times224$. To process RGB and depth data locally for each vehicle, we use ResNet-18 as the encoder to extract feature maps of size $7\times7$. The RGB features from both vehicles are shared and fused through channel-wise concatenation, and the depth features are processed similarly. Then we apply $1\times1$ convolution to reduce the fused feature maps to the original channel dimensions for RGB and depth, respectively. We subsequently apply cross-attention to fuse the RGB and depth feature maps to generate multi-agent multi-modal feature aggregations. These aggregated features are passed through the decoder and upsampled to produce an output map matching the input image size.

**Experimental Results.**
We evaluate performance using the Mean Intersection over Union (mIoU) metric, which quantifies the average overlap between predicted segmentation outputs and ground truth across all classes. We compare the performance of CAML with AML and FCN. In the AML approach, only the ground vehicle operates, with RGB and depth data available during training but only RGB data used for testing. The FCN approach involves only the ground vehicle operating with RGB data for both training and testing.

We present the experimental results in the following Table. CAML demonstrates superior performance in terms of mIoU across both indoor and outdoor environments. Specifically, CAML achieves an improvement of mIoU for 7.4% in indoor scenario and 10.8% in outdoor scenario compared to AML. We also present the qualitative results in **Fig. 7 in the Appendix**. Despite the noisy input image from the ground vehicle, CAML produces predictions that are closest to the ground truth. This performance improvement can be attributed to CAML's multi-agent collaboration, which provides complementary information to enhance data coverage and offers a more comprehensive understanding of the scenes. Additionally, the utilization of auxiliary depth data during training results in more precise segmentation outputs.

**Table 1**: Experimental results of semantic segmentation on real-world dataset CoPeD using aerial-ground vehicles in indoor and outdoor environments. CAML achieves the highest mIoU in both environments.

| **Approach** | **Indoor mIoU (%)** | **Outdoor mIoU (%)** |
|--------------|---------------------|----------------------|
| FCN          | 51.20               | 56.22                |
| AML          | 55.89               | 60.32                |
| CAML         | 60.05               | 66.83                |

---

> ### Author Response · Authors · 2024-11-24
> **Continuing General Response**
>
> **Ablation Studies.**
> In the ablation studies, we explore another variant of CAML called Pre-fusion CAML, applied to the experiment of aerial-ground vehicles collaborative semantic segmentation. However, it is important to note that this variant can be applied to other domains and experiments as well. In this variant, each vehicle first locally extract feature maps of size $7\times7$ for both RGB and depth modalities. Instead of separately fusing the RGB and depth features between the vehicles, we first fuse the feature maps of RGB and depth within each single vehicle using cross-attention. Then we share and merge the fused RGBD features between vehicles via concatenation. We also apply $1\times1 $ convolution to reduce the feature maps to the original channel dimensions. The multi-agent, multi-modal feature aggregations then pass through the decoder. Finally, we obtain the output map by upsampling to match the input image size. The mIoU of the Pre-fusion CAML is similar to that of CAML, achieving 59.16% and 65.78% for indoor and outdoor environments, respectively. Although the fusion order is different, both versions benefit from robust feature aggregation and multi-agent collaboration, which ultimately results in better segmentation performance. For more details, please refer to **Appendix A.2.2**.
>
> **Question 2 (97B9, fWcH): More experimental and training settings, exact modalities used in the experiments.**
>
> In the experiments of collaborative decision-making for CAV, during data collection at each timestamp, the ego vehicle has a maximum of three collaborative vehicles, provided their distance is within a threshold of 150 meters. For each vehicle, both RGB and LiDAR data are used during training, while only RGB data is used during testing in CAML. We have updated this in **Section 5.1 Data Collection.**
>
> In the experiments of collaborative semantic segmentation with aerial-ground vehicles, we use the dataset CoPeD (Zhou et al., 2024), with one aerial vehicle and one ground vehicle, in two different real-world scenarios of the indoor NYUARPL and the outdoor HOUSEA. Additionally, we introduce noise to the RGBD data collected by the ground vehicle. For both aerial and ground vehicles, RGB and depth data are used during training, while only RGB data is used during testing in CAML.
>
> For training, we employ a batch size of 32 and the Adam optimizer with an initial learning rate of $1e{-3}$, and a Cosine Annealing Scheduler to adjust the learning rate over time. The model is trained on an Nvidia RTX 3090 GPU with AMD Ryzen 9 5900 CPU and 32 GB RAM for 200 epochs. We have updated these details in **Appendix A.3.1**. For the comparative training complexity of AML and CAML in both experiments, we list in the following Table 2 and 3.
>
> **Table 1**: Training complexity of AML and CAML in collaborative decision-making for connected autonomous driving.
>
> | **Approach** | **Parameters** | **Time/epoch** |
> |--------------|----------------|----------------|
> | AML          | 19.5M          | 34s            |
> | CAML         | 39.3M          | 73s            |
>
> **Table 2**: Training complexity of AML and CAML in collaborative semantic segmentation for aerial-ground vehicles.
>
> | **Approach** | **Parameters** | **Time/epoch** |
> |--------------|----------------|----------------|
> | AML          | 13.5M          | 3s             |
> | CAML         | 25.5M          | 7s             |
>
> **Question 3 (97B9, fWcH): Limitations and failure case analysis.**
>
> Even though the advancements of CAML, there are some limitations and failure cases. One failure case is that if the modalities are misaligned, the model may struggle to perform effective fusion, leading to incorrect predictions. For instance, if modalities such as RGB and depth are captured at different time intervals or from non-overlapping fields of view, combining them effectively can be difficult. The auxiliary modalities or views from collaborative agents may become noise, useless or even degrading performance. Another limitation is the increasing system complexity. As the number of agents increases, the complexity of the system grows. The fusion of multi-agent and multi-modal data introduces challenges related to coordination overhead, which may lead to delays in the collaborative learning process. We have added this analysis in **Section 6 Conclusions and Limitations**.
>
> [1] Yang Zhou, Long Quang, Carlos Nieto-Granda, and Giuseppe Loianno. Coped-advancing multi-
> robot collaborative perception: A comprehensive dataset in real-world environments. IEEE
> Robotics and Automation Letters, 2024.
>
> [2] Jonathan Long, Evan Shelhamer, and Trevor Darrell. Fully convolutional networks for semantic
> segmentation. In Proceedings of the IEEE conference on computer vision and pattern recognition,
> pp. 3431–3440, 2015.

---

### Meta-Review · Area_Chair_adAD · 2024-12-20

**Metareview:**

(a) Summary: This paper generalizes Auxiliary Modality Learning (AML) to the task of cooperative perception in the context of connected and automated vehicles.
(b) Strengths: The paper is generally well-written and easy to follow. The proposed approach seems reasonable. The experimental results seem to support the authors' claims to some extent.
(c) Weaknesses: The reviewers pointed out a few concerns and issues of the paper, such as lack of baseline comparisons, lack of technical details and explanations, lack of discussions on limitations, and some claims are not fully convincing.
(d) The authors' rebuttal addressed some of the major concerns in clarity. However, there still remain major issues unresolved after the rebuttal, such as the lack of baseline comparison on the real-world dataset. The paper still needs significant further refinement. The majority of reviewers gave a final rating of below borderline.

**Additional Comments On Reviewer Discussion:**

The authors' rebuttal addressed some of the major concerns in clarity. However, there still remain major issues unresolved after the rebuttal, such as the lack of baseline comparison on the real-world dataset.

---

### Decision · Program_Chairs · 2025-01-22

Reject